# Could SP-A and SP-D Serum Levels Predict COVID-19 Severity?

**DOI:** 10.3390/ijms25115620

**Published:** 2024-05-22

**Authors:** Luca Maddaloni, Veronica Zullino, Ginevra Bugani, Alessandro Lazzaro, Matteo Brisciani, Claudio Maria Mastroianni, Letizia Santinelli, Franco Ruberto

**Affiliations:** 1Department of Public Health and Infectious Diseases, Sapienza University of Rome, 00185 Rome, Italy; alessandro.lazzaro@uniroma1.it (A.L.); claudio.mastroianni@uniroma1.it (C.M.M.); letizia.santinelli@uniroma1.it (L.S.); 2Department of General and Specialistic Surgery, Sapienza University of Rome, 00185 Rome, Italy; v.zullino@policlinicoumberto1.it (V.Z.); m.brisciani@policlinicoumberto1.it (M.B.); f.ruberto@policlinicoumberto1.it (F.R.)

**Keywords:** SP-A, SP-D, SARS-CoV-2, collectins, surfactant proteins

## Abstract

Given the various clinical manifestations that characterize Coronavirus Disease 2019 (COVID-19), the scientific community is constantly searching for biomarkers with prognostic value. Surfactant proteins A (SP-A) and D (SP-D) are collectins that play a crucial role in ensuring proper alveolar function and an alteration of their serum levels was reported in several pulmonary diseases characterized by Acute Respiratory Distress Syndrome (ARDS) and pulmonary fibrosis. Considering that such clinical manifestations can also occur during Severe Acute Respiratory Syndrome Coronavirus 2 (SARS-CoV-2) infection, we wondered if these collectins could act as prognostic markers. In this regard, serum levels of SP-A and SP-D were measured by enzyme immunoassay in patients with SARS-CoV-2 infection (n = 51) at admission (T0) and after seven days (T1) and compared with healthy donors (n = 11). SP-D increased in COVID-19 patients compared to healthy controls during the early phases of infection, while a significant reduction was observed at T1. Stratifying SARS-CoV-2 patients according to disease severity, increased serum SP-D levels were observed in severe compared to mild patients. In light of these results, SP-D, but not SP-A, seems to be an eligible marker of COVID-19 pneumonia, and the early detection of SP-D serum levels could be crucial for preventive clinical management.

## 1. Introduction

The continuous global pandemic of Coronavirus Disease 2019 (COVID-19), caused by the Severe Acute Respiratory Syndrome Coronavirus 2 (SARS-CoV-2), represents a public health menace on a worldwide scale [1]. Most SARS-CoV-2-infected patients are asymptomatic or characterized by mild symptoms (throat, loss of taste and smell, malaise and joint pain). However, in older individuals (>50 years old) or with comorbidities (hypertension, diabetes, impaired immunity), the risk of severe manifestations of COVID-19 is higher [2]. The most common clinical manifestation is Acute Respiratory Distress Syndrome (ARDS) with hypoxemic respiratory failure. During this kind of severe critical illness, the overexpression of proinflammatory cytokines leads to endothelial dysfunction and damage. Considering the heterogeneous prognosis of COVID-19 patients, the scientific community has long wondered about potential biomarker candidates for a poorer prognosis. This can enable therapeutic interventions and monitor patients within an appropriate window of action. SARS-CoV-2 has the capacity to affect type I pneumocytes, type II pneumocytes and alveolar macrophages, leading to impaired lung function [3]. Pulmonary surfactants consist primarily of phospholipids and a smaller fraction of surfactant proteins (SP-A, SP-B, SP-C, SP-D) [4]. These surfactant phospholipids, located within the alveolar space, have a critical role in reducing the surface tension along the alveolar walls during inhalation and preventing alveolar collapse after exhalation [5]. In particular, SP-A and SP-D are collectins that interact with the surface oligosaccharides of viruses, and this phenomenon bolsters viral clearance, mostly through the actions of macrophages and monocytes [5]. Additionally, SP-A, as an immunomodulator, has the capacity to restrain dendritic cell maturation and inhibit the excessive release of IL-8 by eosinophils [6]. Notably, both SP-A and SP-D can bind to the spike protein of SARS-CoV-2, inhibiting the virus’s ability to infect host cells [7]. Following SARS-CoV-2 invasion, the ensuing viral replication and damage to type II pneumocytes negatively affect surfactant production, giving rise to breathing difficulties and the development of ARDS in COVID-19 patients. To this extent, in more severe instances, the extensive loss of these critical “alveolar defenders” culminates in respiratory failure [3]. Several studies evaluated SP-D levels in bronchoalveolar lavage fluid (BALF). Decreased SP-D levels were reported in children affected by cystic fibrosis with acute lung infections or in children with Gastroesophageal Reflux Disease (GERD) who have continuous acid airway injury and suffer from chronic lung infections [8,9]. Furthermore, decreased SP-D levels in BALF were observed in patients affected by ARDS with worse oxygenation compared to patients with good oxygenation. In addition, in smokers, BALF SP-D levels are reduced compared to nonsmokers, while increased levels were observed in patients with sarcoidosis, asthma, eosinophilic pneumonia and pulmonary alveolar proteinosis (PAP) [10]. Several studies assessed the value of SP-D serum levels as a disease marker for human lung diseases. This evidence concerns patients with various forms of interstitial lung disease, including interstitial pneumonia with collagen disease, idiopathic pulmonary fibrosis, sarcoidosis and others. The mechanisms that explain how pulmonary SP-D comes into the circulation are unclear, but there are several hypotheses: inflammatory conditions, as in ARDS, induce an increased permeability of lung vessels that might result in alveolar to vascular shift of SP-D; the same inflammatory status could be responsible for a loss of epithelial secretory cells integrity and an efflux of SP-D from epithelial cells into alveolar vessels [10,11,12].

Considering the prognostic increase in SP-A and SP-D serum concentrations in patients with lung damage (idiopathic pulmonary fibrosis and interstitial pneumonia) with collagen vascular diseases [13], we hypothesized that in COVID-19 as well, the levels of these collectins could predict the severity of the disease.

## 2. Results

### 2.1. Study Participants

A total of 51 SARS-CoV-2-infected patients and 11 healthy donors were enrolled in this study. Demographic and clinical characteristics of the study population are reported in Table 1.

### 2.2. Serum SP-A and SP-D Levels Evaluation

At baseline, COVID-19 patients displayed higher median levels of SP-D protein (*p* = 0.0063) compared to healthy donors, while no differences were recorded for SP-A levels between both groups (*p* = 0.0824).

At T1, serum SP-A and SP-D levels were similar among COVID-19 patients and healthy donors (*p* = 0.3451 and *p* = 0.0569, respectively). When comparing T0 and T1, a significant reduction in serum SP-A and SP-D levels was observed at T1 among COVID-19 patients (*p* = 0.0123 for both proteins) (Figure 1).

To investigate the possible association between surfactant protein expression and disease severity, patients were stratified according to respiratory support requirement: the mild group was in ambient air or needed VMK support (n = 21), while patients under CPAP were considered as severe (n = 30) (Table 2). A lower vaccination rate against SARS-CoV-2 was found in the severe compared to the mild patient group. None of these patients developed ARDS. Serum SP-A levels were comparable between healthy controls, mild and severe COVID-19 patients at both time points (*p* > 0.05), while the only difference observed was a longitudinal reduction in the mild group (*p* = 0.0153). Regarding SP-D, no differences were recorded for serum levels of this protein among healthy donors and mild patients at both time points. By contrast, severe patients exhibited higher SP-D levels at T0 (*p* = 0.0006) and T1 (*p* = 0.0063) compared to the control group. In addition, the serum levels of this protein were higher in the COVID-19 severe group compared to the mild one at T1 (*p* = 0.0369) (Figure 2).

## 3. Discussion

Several studies associated SARS-CoV-2 infection with pulmonary fibrosis and ARDS [14,15]. Previous evidence from the literature in a plethora of pulmonary diseases has reported that a decrease in SP-A concentration in BALF was associated with ARDS, while both SP-A and SP-D concentrations were reduced in BALF fluids of patients with pulmonary fibrosis. By contrast, their serum concentration increased in these patients [13,16,17]. Afterward, elevated plasma SP-D levels were observed in patients experiencing pneumonia after SARS-CoV infection [18], and more recently, high serum levels of this protein have been associated with severe COVID-19, suggesting its potential utility for the early identification of patients who can face worsening conditions [12,19]. In light of these considerations, we measured the levels of these two collectins in patients’ serum and evaluated their association with the clinical outcome. SP-D, but not SP-A, increased among COVID-19 patients compared to healthy controls during the early phases of infection, while a significant reduction was observed at T1. To better clarify the role of the two soluble lectin members family, we stratified SARS-CoV-2 patients according to mild and severe disease, and we observed increased serum levels of SP-D in patients under CPAP compared to patients supported with VMK or without respiratory support at T1. This may be due to an earlier recovery of the latter than the former, as they are presumably characterized by less extended lung injury. During COVID-19, a previous SARS-CoV-2 vaccination plays a relevant role in the fight against infection. In line with this, our group of severe patients requiring more invasive respiratory support was characterized by a higher frequency of unvaccinated individuals. Our findings suggest a role for SP-D as a predictive marker of COVID-19 outcome, as proposed in other studies, in which this collectin levels were associated with ARDS severity and clinical outcome [11,20,21]. On the other hand, no differences were observed for SP-A serum levels according to COVID-19 severity, as also previously observed by Takenaka et al. [22] However, a higher frequency of vaccination was observed in the mild group than in the severe group, emphasizing the importance of vaccination in reducing COVID-19 severity [23]. The lower validity of SP-A as a marker might be due to its reduced serum levels compared to SP-D, both under physiological and pathological conditions. Despite these findings, the limitations of this study are the small sample size, significant age and vaccination differences between COVID-19 patients and controls and the lack of quantification of these proteins’ levels in BALF to assess their likely reduction corresponding to their serum increase. Nevertheless, surfactant protein levels do not appear to depend on age or gender but rather on smoking habits or pathological conditions in the lungs [24,25]. Further studies might focus on the correlation between such collectins and more objective clinical indicators, such as lung imaging and haemogasanalysis parameters. In conclusion, our study suggests SP-D, but not SP-A, as an eligible marker of COVID-19 pneumonia, and the early detection of SP-D serum levels could be a determinant for a preventive treatment of COVID-19.

## 4. Materials and Methods

### 4.1. Study Participants and Ethical Approvals

Peripheral blood samples were consecutively collected at T0 (at admission) and T1 (seven days post hospitalization) from SARS-CoV-2 infected individuals hospitalized at the Policlinico Umberto I, Sapienza University of Rome (Italy) from January to March 2022. A control group of healthy donors was also enrolled. Patients who were ≥18 years old, who had received vaccination or were not vaccinated, with a positive SARS-CoV-2 RT-PCR test were enrolled. Patients with any one of the following conditions were excluded: a diagnosis of autoimmune lung disease; pulmonary fibrosis; obstructive pulmonary disease; or bacterial pneumonia. Oxygen therapy was delivered via a Venturi mask (VMK) in spontaneous breathing patients. If hypoxemia persisted, continuous positive airway pressure (CPAP) was applied. Of the 51 patients studied, 11 patients had comorbidities (21.6%) and 40 had no comorbidities (78.4%). The most common comorbidities were arterial hypertension (37.3%), cardiovascular diseases (25.4%), oncological diseases (17.6%) and controlled lung diseases (5.1%). Among patients with controlled lung diseases, 4.08% had chronic obstructive pulmonary disease (COPD) and 1.02% had pulmonary emphysema. In all cases, patients were under treatment with oral steroids. Hematologic diseases were the most frequent pathological condition in the field of oncological diseases. Furthermore, we used body mass index to identify patients with obesity: 8.16% had an obesity class I, while the other had a body mass index of less than 30 kg/m^2^. Finally, patients or healthy donors with a smoking habit were not enrolled.

The COVID-19 case definitions of the European Centre for Disease Prevention and Control (ECDC) were adopted [26], and the severity stratification was based on World Health Organization (WHO) criteria [27]. The study was approved by the ethics committee of the Policlinico Umberto I Hospital, “Sapienza” University of Rome, and informed consent was obtained from both COVID-19 patients and healthy individuals. Finally, patients’ data were anonymized, and a progressive alphanumeric code was assigned to them.

### 4.2. Serological Evaluation of SP-A and SP-D Levels

Twenty milliliters of whole blood were collected; serum was separated by centrifugation at 1620 RCF for 10′ within one hour after collection and stored at −80 °C for further analysis. Serum SP-A and SP-D levels at T0 and T1 were evaluated using a commercially available enzyme-linked immunosorbent assay (ELISA) kit [Catalog numbers MBS167144 (MyBioSource, San Diego, CA, USA) and ELH-SPD (RayBiotech Life, Peachtree Corners, GA, USA), respectively], according to the manufacturer’s protocol. The assay was based on an indirect assay with an antigen–antiserum bound and a conjugate antibody using HRP-Streptavidin as a substrate for the detection. Absorptions were measured at 450 nm with a Microplate Multimode Reader (Bio-Rad Laboratories, Hercules, CA, USA). Concentrations in each sample were calculated from the standard curve, obtained by plotting absorbance versus known standard concentrations. SP-A and SP-D concentrations were expressed as ng/L and ng/mL, respectively.

### 4.3. RT-qPCR Detection of SARS-CoV-2 RNA

Viral RNA was extracted from nasopharyngeal swabs using a Versant SP 1.0 Kit (Siemens Healthcare Diagnostics, Milan, Italy). Briefly, 10 μL of extracted RNA was reverse-transcribed and simultaneously amplified by a real-time RT-PCR system (RealStar SARS-CoV-2 RT-PCR, Altona Diagnostics, Hamburg, Germany), targeting E and S viral genes.

### 4.4. Statistical Analysis

Patients’ data were expressed as median (interquartile ranges) or number (percentage). Demographic and clinical patients’ characteristics were analyzed using the “N-1” Chi-squared test. Cross-sectional data between patients and healthy donors were analyzed using the Mann–Whitney *U* test, while the Wilcoxon signed–rank test for paired samples was used to evaluate longitudinal data between T0 and T1. A *p*-value of less than 0.05 was considered statistically significant. Statistical analyses were performed using GraphPad Prism software, version 9.4 (GraphPad Software Inc., La Jolla, CA, USA).

## Figures and Tables

**Figure 1 ijms-25-05620-f001:**
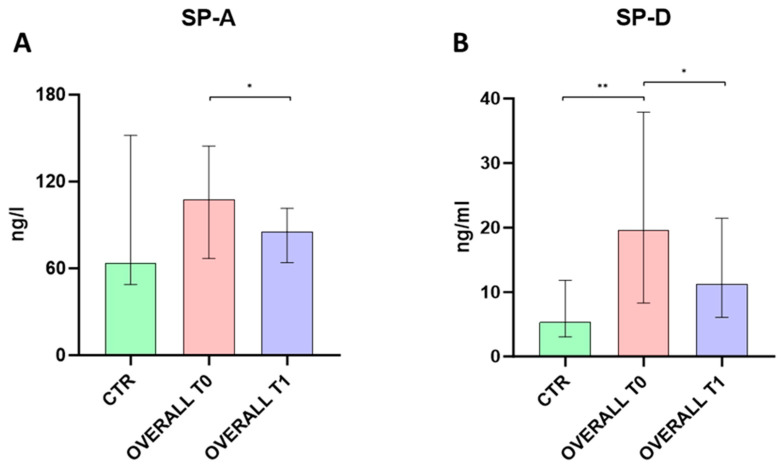
Comparison of serum SP-A (**A**) and SP-D (**B**) between SARS-CoV-2 infected patients at hospitalization (T0) and 7 days after (T1) and healthy controls. Data were analyzed using the Mann–Whitney *U*-test for unpaired samples, and the Wilcoxon signed–rank test for paired samples. * *p* < 0.05; ** *p* < 0.01.

**Figure 2 ijms-25-05620-f002:**
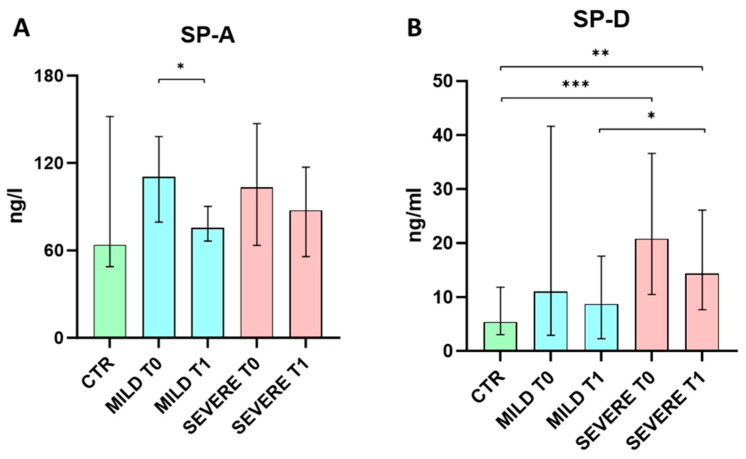
Comparison of serum SP-A (**A**) and SP-D (**B**) between mild and severe SARS-CoV-2 infected patients at hospitalization (T0) and 7 days after (T1) and healthy controls. Data were analyzed using the Mann–Whitney U-test for unpaired samples, and the Wilcoxon signed–rank test for paired samples. * *p* < 0.05; ** *p* < 0.01, *** *p* < 0.001.

**Table 1 ijms-25-05620-t001:** Demographic and clinical characteristics of SARS-CoV-2 infected patients and healthy donors.

Parameters	SARS-CoV-2 Infected Patients (n = 51) (A)	Healthy Donors (n = 11) (B)	A vs. B *p*-Values
Age (years) ^a^	68 (56.5–78)	39 (30–52)	<0.0001
Male ^b^	24 (47%)	6 (54.5%)	0.7465
SARS-CoV-2 vaccination (at least 2 doses) ^b^	14 (27%)	11 (100%)	<0.0001
Respiratory support ^b^	VMK 12 (23%) A.A. 9 (18%) CPAP 30 (59%)	NA	-
D-dimer (μg/L) ^a^	1092 (631.5–1621)	NA	-
PCR (mg/dL) ^a^	4.17 (1.39–6.5)	NA	-
Ferritin (ng/mL) ^a^	823 (357–1244)	NA	-
LDH (Ui/L) ^a^	242 (209.5–252)	NA	-
Fibrinogen (mg/dL) ^a^	407 (325–440)	NA	-
Lymphocytes (10^3^/mL) ^a^	1285 (815–1795)	NA	-

NA = not applicable, VMK = Venturi Mask, CPAP = Continuous Positive Airway Pressure, A.A. = Ambient Air. ^a^ Data are expressed as median (interquartile ranges). ^b^ Data are expressed as numbers (percentages). Differences in demographic characteristics between SARS-CoV-2 infected patients at admission (T0) and healthy individuals were evaluated using Mann–Whitney *U* and Chi-squared tests. *p* < 0.05 was considered statistically significant.

**Table 2 ijms-25-05620-t002:** Demographic and clinical characteristics of mild and severe SARS-CoV-2 infected patients.

Parameters	Mild SARS-CoV-2 Infected Patients (n = 21) (A)	Severe SARS-CoV-2 Infected Patients (n = 30) (B)	A vs. B *p*-Values
Age (years) ^a^	75 (62–83)	65 (56–74)	0.0739
Male ^b^	8 (38%)	66 (53%)	0.2955
SARS-CoV-2 vaccination (at least 2 doses) ^b^	12 (57%)	2 (7%)	0.0001
Respiratory support ^b^	VMK 12 (57%) A.A. 9 (43%)	CPAP 30 (100%)	0.0001
ARDS development ^b^	0 (0%)	0 (0%)	-
D-dimer (μg/L) ^a^	1209 (669–1621)	1007 (535–1920)	0.6906
PCR (mg/dL) ^a^	4.63 (0.83–6.5)	4.17 (1.4–7.35)	0.6814
Ferritin (ng/mL) ^a^	821 (239–1199)	823 (511–1294)	0.5183
LDH (Ui/L) ^a^	284 (199–390)	265 (229–284)	0.9007
Fibrinogen (mg/dL) ^a^	514 (384–555)	444 (362–555)	0.2244
Lymphocytes (10^3^/mL) ^a^	910 (630–1345)	1055 (763–1465)	0.1622

VMK = Venturi Mask, CPAP = Continuous Positive Airway Pressure, A.A. = Ambient Air. ^a^ Data are expressed as median (interquartile ranges). ^b^ Data are expressed as numbers (percentages). Differences in demographic characteristics between mild and severe SARS-CoV-2 infected patients at admission (T0) were evaluated using Mann–Whitney *U* and Chi-squared tests. *p* < 0.05 was considered statistically significant.

## Data Availability

The data that support the findings of this study are available from the corresponding authors, L.M. and G.B., upon reasonable request.

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
