# Peer review of "Could SP-A and SP-D Serum Levels Predict COVID-19 Severity?"

_ijms, 2024, doi:10.3390/ijms25115620_

Round 1
Reviewer 1 Report
Comments and Suggestions for Authors
The manuscript explores the potential of surfactant proteins A (SP-A) and D (SP-D) as biomarkers for predicting COVID-19 severity. The authors measured serum levels of SP-A and SP-D in SARS-CoV-2-infected patients at admission (T0) and after 7 days (T1), comparing them with healthy controls. Key findings include higher serum SP-D levels in COVID-19 patients at T0, which decreased at T1. Severe COVID-19 patients had higher SP-D levels at T1 compared to mild cases. No significant differences were observed for SP-A levels. The authors suggest SP-D as a potential prognostic marker for COVID-19 pneumonia. Overall, the study's topic is intriguing and could be further improved by addressing the following suggestions:
- Improve the introduction by providing more background on SP-A and SP-D roles in lung function and COVID-19 pathogenesis.
- Clarify criteria for stratifying patients into mild and severe groups, and provide details on respiratory support methods.
- Expand the discussion to better contextualize findings within existing literature and address limitations or alternative interpretations.
- Make minor grammatical and language edits for clarity and readability.
Make minor grammatical and language edits for clarity and readability
Author Response
|
Dear Reviewer, Thank you very much for taking the time to review this manuscript and for your precious comments. Please find the detailed responses below and the corresponding revisions highlighted in the re-submitted files.
Comments1: Improve the introduction by providing more background on SP-A and SP-D roles in lung function and COVID-19 pathogenesis.
Response 1: we tried to improve the introduction section, especially including more background regarding surfactant proteins role in COVID-19 pathogenesis.
Comments 2: Clarify criteria for stratifying patients into mild and severe groups, and provide details on respiratory support methods.
Response 2: we tried to make stratification clearer. Regarding respiratory support methods details we think it is well described in the “Study participants and ethical approvals” sub-section in “Materials and methods”.
Comments 3: Expand the discussion to better contextualize findings within existing literature and address limitations or alternative interpretations.
Response 3: we tried to enrich the discussion section to make it a bit clearer.
Comments 4: Make minor grammatical and language edits for clarity and readability.
Response 4: Thanks, we edited some grammatical and language errors.
|
Reviewer 2 Report
Comments and Suggestions for Authors
Manuscript Number: ijms-2992052
I have thoroughly reviewed the communication entitled "Could SP-A and SP-D serum levels predict COVID-19 severity?". This communication focuses on the study if the serum levels of surfactant protein A (SP-A) and D (SP-D) are biomarkers of COVID-19 severity. I have some concerns regarding the limitations mentioned by the authors and the need for further clarification on certain aspects.
The authors acknowledge two important limitations of their work: small sample size and significant age differences between groups. Given these limitations, it is crucial to determine whether differences in SP-A and SP-D levels across age groups or between sexes are described in the literature. Understanding potential variations in these protein levels based on age and sex could provide valuable context for interpreting the study's findings.
Additionally, I am surprised by the small size of the healthy control group and the significant age disparity compared to the other group. Given these limitations, I wonder if it would be possible to substitute the healthy control group with a larger group of individuals of similar ages to the other group. The manuscript does not provide details on how these healthy individuals were recruited, whether smokers were excluded, or other relevant factors. Clarifying these aspects would enhance the transparency and reliability of the study.
In the abstract, collectins are mentioned in reference to SP-A and SP-D, without prior mention that these two proteins belong to this family of proteins.
The sentence “However, in individuals older (>50 years old) or with comorbidities (hypertension, diabetes, impaired immunity) the risk of severe manifestations of COVID-19 is higher.” needs a proper bibliographic reference.
Comments on the Quality of English LanguageModerate editing of English language is required.
Author Response
Dear Reviewer,
Thank you very much for taking the time to review this manuscript and for your precious comments. Please find the detailed responses below and the corresponding revisions highlighted in the re-submitted files.
Comments1: The authors acknowledge two important limitations of their work: small sample size and significant age differences between groups. Given these limitations, it is crucial to determine whether differences in SP-A and SP-D levels across age groups or between sexes are described in the literature. Understanding potential variations in these protein levels based on age and sex could provide valuable context for interpreting the study's findings.
Additionally, I am surprised by the small size of the healthy control group and the significant age disparity compared to the other group. Given these limitations, I wonder if it would be possible to substitute the healthy control group with a larger group of individuals of similar ages to the other group. The manuscript does not provide details on how these healthy individuals were recruited, whether smokers were excluded, or other relevant factors. Clarifying these aspects would enhance the transparency and reliability of the study.
In the abstract, collectins are mentioned in reference to SP-A and SP-D, without prior mention that these two proteins belong to this family of proteins.
The sentence “However, in individuals older (>50 years old) or with comorbidities (hypertension, diabetes, impaired immunity) the risk of severe manifestations of COVID-19 is higher.” needs a proper bibliographic reference.
Response 1: thanks for the suggestions. We fixed the two errors indicated at the end of the comment. Regarding the healthy control group we added more details about smoking habits. We also reported a reference indicating that the age of patients is not really important in respect of serum surfactant protein levels.
Round 2
Reviewer 2 Report
Comments and Suggestions for Authors
I still have doubts about the healthy donors group.